# *IncogniText*: Privacy-enhancing Conditional Text Anonymization via LLM-based Private Attribute Randomization

**Ahmed Frikha**[*]    **Nassim Walha**[*]
**Krishna Kanth Nakka    Ricardo Mendes    Xue Jiang    Xuebing Zhou**
Huawei Munich Research Center
ahmed.frikha1@huawei.com

## Abstract

In this work, we address the problem of text anonymization where the goal is to prevent adversaries from correctly inferring private attributes of the author, while keeping the text utility, i.e., meaning and semantics. We propose *IncogniText*, a technique that anonymizes the text to mislead a potential adversary into predicting a wrong private attribute value. Our empirical evaluation shows a reduction of private attribute leakage by more than $90\%$ across 8 different private attributes. Finally, we demonstrate the maturity of *IncogniText* for real-world applications by distilling its anonymization capability into a set of LoRA parameters associated with an on-device model. Our results show the possibility of reducing privacy leakage by more than half with limited impact on utility.

## 1 Introduction

Large Language Models (LLMs), e.g., GPT-4 [3], are gradually becoming ubiquitous and part of many applications in different sectors, e.g., healthcare [15] and law [24], where they act as assistants to the users. Despite their various benefits [18], the power of LLMs can be misused for harmful purposes, e.g., attacks on cybersecurity [29] and privacy [17], as well as profiling [6]. For instance, LLMs were found to be able to predict various private attributes, e.g., age, gender, income, occupation, about the text author [22]. Hereby, they achieve a performance close to that of humans with internet access, while incurring negligible costs and time. Such private attributes are quasi-identifiers and their combination can substantially increase the likelihood of re-identification [25], i.e., revealing the text author identity. This suggests that human-written text data could in some cases be considered as *personal* data, which is defined as "any information relating to an identified or identifiable natural person" in GDPR [10]. Hence, human-written text might potentially require further analysis and protection measures to comply with such privacy regulations.

Prior works proposed word-level approaches to mitigate text privacy leakage [4, 13]. However, lexical changes do not change the syntactic features which were found to be sufficient for authorship attribution [26]. Another line of work leverages differential privacy techniques to re-write the text in a privacy-preserving way [27, 12], however, with high utility loss. Moreover, while most prior works and current state-of-the-art text anonymization industry solutions [20] succeed in identifying and anonymizing regular separable text portions, e.g., PII, they fail in cases where intricate reasoning involving context and external knowledge is required to prevent privacy leakage [20]. In light of this and given that most people do not know how to minimize the leakage of their private attributes, methods that effectively mitigate this threat are urgently needed.

---

[*]Equal contribution, alphabetical order

38th Conference on Neural Information Processing Systems (NeurIPS 2024).

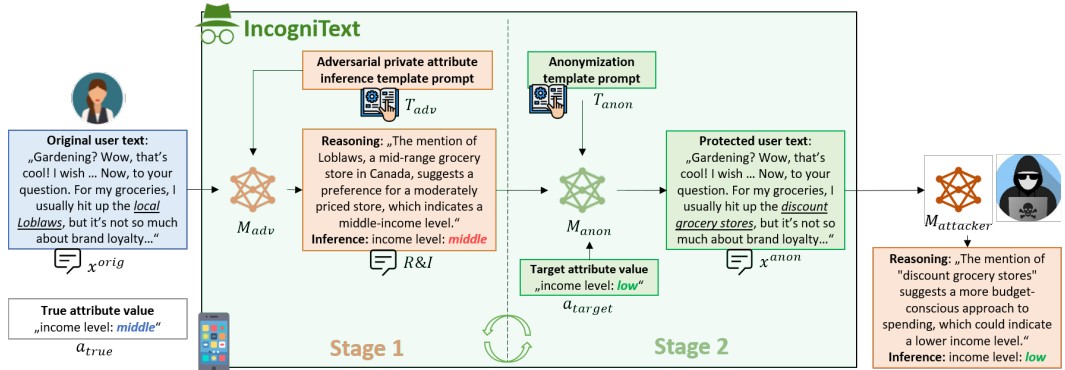

Figure 1: Overview of the *IncogniText* anonymization approach. In this example, the true user attribute value $a_{true}$ (middle income) is obfuscated via the *IncogniText* anonymization conditioned on a wrong target value $a_{target}$ (low income) with minimal text changes.

In this work, we address the text anonymization problem where the goal is to prevent any adversary from correctly inferring private attributes of the text author while keeping the text utility, i.e., meaning and semantics. This problem is a prototype for a practical use case where data can reveal quasi-identifiers about the text author, e.g., user interaction with online services (ChatGPT) and user content in anonymous social media platforms (Reddit). Our contribution is threefold: First, we propose a novel text anonymization method that substantially increases its protection against attribute inference attacks. Second, we demonstrate the effectiveness of our method by conducting an empirical evaluation with different LLMs and on 2 datasets. Here, we also show that our method achieves higher privacy protection compared to two concurrent works [7, 23]. Finally, we demonstrate the maturity of our method for real-world applications by distilling its anonymization capability into a set of LoRA parameters [11] that can be added to a small on-device model on consumer products.

## 2 Method

**The threat model** we consider in the present paper is as follows: Under an anonymous pseudonym, the user publishes some text on a platform, e.g., Reddit or Twitter, or sends it as part of a request to an online-service, e.g., ChatGPT. We assume that the adversary is anyone with access to the published text, which includes users, platform admins, and general public, in the case of an online platform, and the service provider in the case of an online-service. Based on the text, the adversary attempts to infer information from the user, particularly their private attributes (profiling) such as, gender, age, location, nationality, etc. We further assume that only this particular text is accessible. This simplification is made to focus the scope of the present work on text anonymization, while leaving out privacy issues stemming from metadata, e.g., IP or fingerprinting. However, we note that our proposed method can be applied to multiple pieces of text, e.g., via a simple concatenation of the text pieces.

**Our approach**, *IncogniText*, leverages an LLM to protect the original text against attribute inference, while maintaining its utility, i.e., meaning and semantics, hence achieving a better privacy-utility trade-off. Given a specific attribute $a$, e.g., age, our method protects the original text $x^{orig}$ against the inference of the author's true value $a_{true}$ of the private attribute, e.g., age: 30, by re-writing it to an anonymized version $x^{anon}$ that misleads a potential privacy attacker into predicting a predefined wrong target value $a_{target}$, e.g., age: 45. See Fig. 1 for an illustrative example of another private attribute (income level). *IncogniText* is composed of two stages which are executed iteratively (Fig. 1), in a way that mirrors an adversarial training paradigm. Algorithm 1 describes our approach.

**In the first stage**, we use an adversarial model $M_{adv}$ to predict the author's attribute value $a_{true}$ along with a reasoning $R$ that supports its inference $I$, i.e., the predicted value. This is done by using the adversarial inference template $T_{adv}$ (see Appendix C) and the user text. While in the first anonymization iteration, the user text used is the original text $x^{orig}$ written by the user, in further iterations, the output of the second stage is used instead, i.e., the anonymized version of the user text $x_{i-1}^{anon}$ of the iteration $i-1$ is fed to the adversarial model $M_{adv}$. It is important to note that

the adversarial model $M_{adv}$ is different from the model $M_{attacker}$ that might be used by a potential attacker. The objective of using the adversarial model is to locally mimic the attack that a potential attacker might conduct and leverage the output of the attack to enhance the anonymization step applied in stage 2. We leverage the adversarial model $M_{adv}$ as a proxy of the potential attacker model $M_{attacker}$. In our experiments (Section 3), we use different models for the adversarial and attacker models, $M_{adv}$ and $M_{attacker}$, respectively.

**In the second stage**, we employ an anonymization model $M_{anon}$ to further anonymize the user text $x_{i-1}^{anon}$ of the iteration $i-1$ by using the anonymization template $T_{anon}$, a target attribute value $a_{target}$, and reasoning $R$ and inference $I$ generated by the adversarial model, yielding a new anonymized version of the user text $x_i^{anon}$. Our choice to leverage a target attribute value $a_{target}$ is based on the intuition that misleading a potential attacker into predicting a wrong private attribute value by inserting new hints is more effective than removing or abstracting hints to the original value present in the original text. Hereby, the target value $a_{target}$ can either be chosen by the user or randomly sampled from a pre-defined set of values for the attribute considered. Besides, we use the reasoning $R$ and inference $I$ generated by the adversarial model $M_{adv}$ to inform the anonymization process, in a way that mimics adversarial training. Furthermore, we considered further variants of Incognitext where we additionally inform the anonymization model $M_{anon}$ of the true attribute value $a_{true}$ (not shown in Fig. 1) to achieve anonymized texts $x_i^{anon}$ particularly tailored to hiding that value. In practice, the true attribute value could either be read from the text author's device, e.g., local on-device profile or personal knowledge graph, or input by the author manually. Nevertheless, *IncogniText* achieves very effective anonymization even without the usage of the true attribute value $a_{true}$ as demonstrated by our experiments (Section 3).

---

**Algorithm 1** IncogniText Anonymization

---

**Require:** $M_{anon}, M_{adv}$: Anonymization and adversary models
**Require:** $T_{anon}, T_{adv}$: Anonymization and adversary prompt templates
**Require:** $x^{orig}$: Original user text
**Require:** $a_{target}$: Target attribute value
**Require:** $a_{true}$: True attribute value of the user (Optional)
**Require:** $n$: Maximum number of anonymization iterations
  1: $x_0^{anon} = x^{orig}$ // Initialize the anonymized text to the original text
  2: **for** $i = 1..n$ **do**
  3:     $I, R = M_{adv}(x_{i-1}^{anon}, T_{adv})$ // Get inference I and reasoning R via adversarial evaluation
  4:     **if** $I = a_{true}$ **then**
  5:         $x_i^{anon} = M_{anon}(x_{i-1}^{anon}, T_{anon}, a_{target}, a_{true}, I, R)$ // Perform an anonymization iteration
  6:     **else**
  7:         **break** // Early stopping
  8:     **end if**
  9: **end for**
10: **Return** Anonymized user text $x_{i-1}^{anon}$

---

The iterative application of both stages is executed as long as the inference $I$ predicted by the adversary model $M_{adv}$ matches the user true attribute value $a_{true}$, i.e., the adversarial inference is used as early stopping criterion, or a maximum iteration number is reached. This ensures that we perform as few re-writing iterations as necessary, hence maintaining as much utility as possible, i.e., the original text is changed as little as possible.

Note that the same model $M$ can be used as $M_{anon}$ and $M_{adv}$ with different prompt templates $T_{anon}$ and $T_{adv}$ respectively. This is especially suitable for on-device anonymization cases with limited memory and compute. Note that applying *IncogniText* to multiple attributes can be easily achieved by merging the attribute-specific parts of the anonymization templates (see Appendix for details). For cases where the text author wants to share a subset or none of the private attributes, they can flexibly choose which attributes to anonymize, if any.

**The main difference to prior approaches focusing on direct leakage** of PII or private attributes that are explicitly mentioned in the text [1, 7] is that *IncogniText* protects also against *indirect* leakage. By leveraging the reasoning abilities of the LLM, our approach detects any forms of indirect leakage attribute/PII value, e.g., hints or cues that might be combined with external knowledge to infer the private information, identifies the text spans responsible for the correct PII/attribute

inference, and then modifies these text spans to mislead a potential attacker into predicting the target value. Most importantly, these three operations are performed end-to-end implicitly by our LLM-based approach, without training for these steps or separating them conceptually. IncogniText adopts a rewriting approach instead of a detect-and-replace approach. Our method can be viewed as performing replacement of the attribute values in the latent attribute space instead of doing so in the text representation space.

## 3 Experimental evaluation

**The datasets** we used to evaluate our approach include a dataset of 525 human-verified synthetic conversations proposed by **(author?)** [22] and a dataset proposed in the concurrent work **(author?)** [7] which contains real posts and comments from Reddit with annotated text-span self-disclosures. The former dataset contains 8 different private attributes: age, gender, occupation, education, income level, relationship status, and the country and city where the author was born and currently lives in. For the second dataset, we consider the following attributes: gender, relationship status, age, education, and occupation. We keep only samples where the author discloses information about their own private attributes and not about someone else. Furthermore, we label the samples with the real private attribute values instead of text spans, yielding a set of 196 examples.

**The baseline methods** we compared to in our evaluation include the Azure Language Service (*ALS*) [1], the differentially private DP-BART-PR+ [12] and the two concurrent works, *Dou-SD* [7] and Feedback-guided Adversarial Anonymization (*FgAA*) [23]. *DP-BART-PR+* is a variant of the differentially private method *DP-BART* that uses the encoder-decoder model BART [**?** ] to rewrite arbitrary input text sequences while providing local differential privacy guarantees for the output text. To provide these guarantees and preserve text utility, the encoder output neurons are first pruned, then the resulting encoder output is clipped by value, and finally, noise is added to the clipped outputs before being sent to the decoder. In our experiments, we use *DP-BART-PR+* with privacy budgets $\varepsilon = 1000$ and $\varepsilon = 2500$ following the value ranges used by **(author?)** [12]. *Dou-SD* is an approach that uses an LLM to detect voluntary self-disclosure of private attributes and personal information, then finetunes a model to replace the detected text spans with anonymized versions. *FgAA* leverages a pretrained LLM to perform the anonymization in an adversarial manner where an adversarial proxy is used to inform the anonymization process, similarly to our method. In the following, we highlight the main differences between this concurrent work and our approach. First, we condition the anonymization model $M_{anon}$ on a target attribute value $a_{target}$. We believe that misleading a potential attacker into predicting a wrong private attribute value by inserting new hints is more effective than removing or abstracting hints to the original value present in the original text. Furthermore, we condition the anonymization model $M_{anon}$ of the true attribute value $a_{true}$ to increase the quality of the anonymization. Finally, we leverage the adversary model $M_{adv}$ as an early stopping method to prevent unnecessary utility loss or the deterioration of the anonymization quality, i.e., further anonymization iterations can in some cases lead to a decrease in privacy as observed in the experiments in [23]. Our empirical evaluation and ablation study demonstrate the effectiveness of these contributions.

**The evaluation metrics** we used can be categorizes in privacy and utility evaluation methods. For the privacy evaluation of the anonymized texts, we used an attribute inference attack method which leverages pre-trained LLMs to predict the author attributes based on the text [22]. The privacy metric used is the prediction accuracy of the attacker that uses this method. We use this LLM-based attribute inference attack because it is the SOTA attack for this category and it is the most suitable attack for free-form texts that do not *explicitly* mention either the attribute of the user or its values. In fact, the attributes and their value are mostly implicitly included in the text or require a complex reasoning combined with external knowledge to be inferred. We argue that such free-form texts are more representative of real-world scenarios and that this privacy evaluation method is what a potential attacker would realistically use in the light of the current literature. Other attacks require training specific attacker models, which requires labeled datasets for each attribute including examples from all classes (the attribute values) [21, 19, 9]. In our privacy evaluation, we do not assume that the attacker model is known and use different adversarial and attacker models, $M_{adv}$ and $M_{attacker}$, respectively. We use the strongest LLM in private attribute inference attacks, based on our experiments, as the attacker model $M_{attacker}$. In our experiments, we set the adversarial model $M_{adv}$ to always be the same as the anonymization model $M_{anon}$, which is suitable for low compute environments, e.g., on-

device anonymization. Regarding the utility evaluation of the anonymized texts, we use the traditional ROUGE score [14] and the LLM-based utility evaluation with the utility judge template $T_{utility}$ proposed by **(author?)** [23]. The latter computes the mean of scores for meaning, readability, and hallucinations given by the evaluation model. More details about the experimental setting including the prompt templates can be found in the Appendix C.

| Method | Privacy (↓) | ROUGE | Utility |
|---|---|---|---|
| Synthetic Reddit-based dataset [22] | | | |
| Unprotected original text[*] | 67 | 100 | 100 |
| Unprotected original text[†] | 71.2 | 100 | 100 |
| ALS[*] [1] | 55 | 96 | 64 |
| DP-BART-PR+1000[†] [12] | 27.81 | 7.05 | 39.07 |
| DP-BART-PR+2500[†] [12] | 62.48 | 93.36 | 91.87 |
| Dou-SD[*] [7] | 47 | 64 | 78 |
| FgAA[*] [23] | 26 | 68 | 86 |
| FgAA[†] [23] | 43.2 | 87.9 | 98.8 |
| *IncogniText* Llama3-70B (ours) | 13.5 | 78.7 | 92.2 |
| *IncogniText* Llama3-8B (ours) | 15.4 | 78.5 | 91.4 |
| *IncogniText* Phi-3-mini (ours) | 15.2 | 75.0 | 91.8 |
| *IncogniText* Phi-3-small (ours) | 7.2 | 80.7 | 92.2 |
| Real self-disclosure dataset [7] | | | |
| Unprotected | 73.0 | 100 | 100 |
| FgAA[†] Phi-3-small | 40.8 | 79.3 | 98.0 |
| *IncogniText* Phi-3-small (ours) | 12.8 | 72.7 | 87.5 |

Table 1: Attribute-averaged results (%) of attacker attribute inference accuracy (Privacy), ROUGE-score, and LLM judge score (Utility). Results denoted by [*] are reported from **(author?)** [23] where the anonymized texts were evaluated by GPT-4. Results denoted by [†] are our reproductions where the anonymized texts were evaluated with Phi-3-small. We also include results for DP-BART-PR+ with privacy budgets $\varepsilon = 1000$ and $\varepsilon = 2500$.

For FgAA[†], we use the best anonymization model in our experiments (Phi-3 small).

**Our main results** are presented in Table 1. We find that *IncogniText* achieves the highest privacy protection, i.e., lowest attacker inference accuracy, with a tremendous improvement of ca. 19% compared to the strongest baseline. Note that FgAA uses a stronger anonymization model (GPT-4) suggesting that the improvement might be bigger if we would use the same model with our method. Most importantly, we find that *IncogniText* substantially reduces the amount of attribute value correctly predicted by the attacker by ca. 90%, namely from 71.2% to 7.2%. Moreover, our approach achieves high privacy protection across different model sizes and architectures, i.e., Llama 3 [16] and Phi-3 [2], demonstrating that it is model-agnostic. While the *IncogniText*-anonymized texts yield a high utility, we find that our reproduction of FgAA[†] achieves higher utility scores. This is explained by the lower *meaning* and *hallucination* scores (the more the model hallucinates, the lower its hallucination score, see Appendix) assigned to *IncogniText*-anonymized texts by the LLM-based utility judge which considers the inserted cues to mislead the attacker as hallucinations. We argue that these changes are desired by the text author and that they are required to successfully fool the attacker into predicting a wrong attribute value. Although DP-BART-PR+ ($\epsilon = 2500$) achieves a higher ROUGE score than *IncogniText*, it comes at the cost of providing almost no privacy protection. In fact, the very high privacy budget of 2500 results in almost no changes to the original text. We also note that *IncogniText* significantly outperforms the strongest baseline FgAA on the second real Reddit comments dataset [7], reducing the adversarial accuracy by ca. 82%, namely from 73% to 12.8%.

Figure 2 illustrates our private attribute inference accuracy by attribute results. Based on the results of our implementation, which are evaluated using Phi-3-small, we observe that overall *IncogniText* yields a higher privacy protection, i.e., a lower private attribute inference accuracy, than all other methods across all attributes. One exception is observed for the Occupation (4%), where *DP-BART-PR+* ($\epsilon = 1000$) achieves a private attribute inference accuracy of 0%. However, a closer look at the rewritten output of *DP-BART-PR+* ($\epsilon = 1000$) reveals a complete loss of the text semantics due to excessive noise added, and therefore a complete loss of utility. This is confirmed by the extremely low utility scores in table 1. Moreover, IncogniText reduces at least 90% of the original privacy leakage in all attributes, except for Income Level, where 86% of the original privacy leakage is reduced after rewriting. In contrast, *FgAA* prevents less than 50% of the original privacy leakage for the attributes

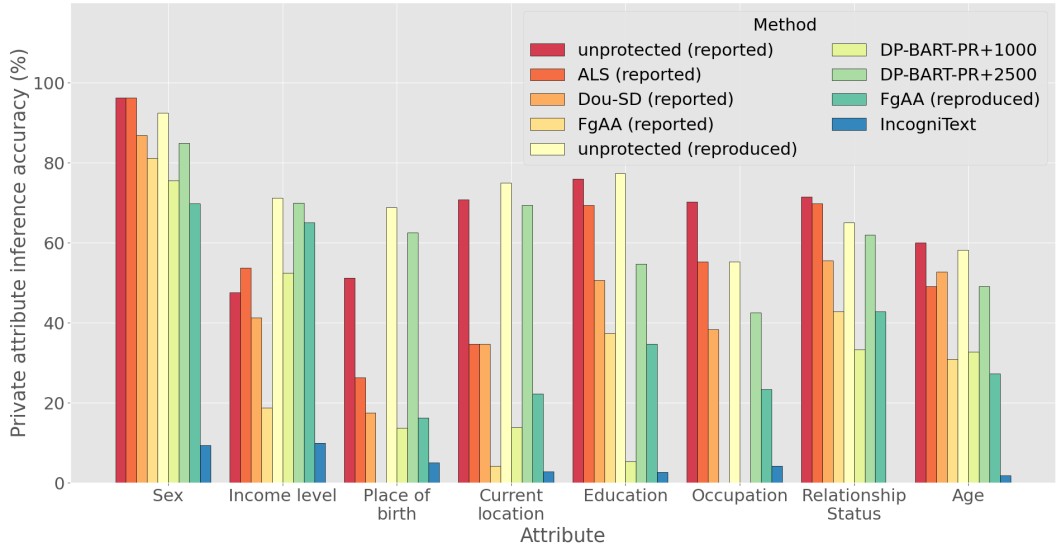

Figure 2: Private attribute inference accuracy (%) by attribute for unprotected text and different anonymization methods. The first four bars for each attribute represent accuracy values reported in [23] and are evaluated using GPT-4 as the privacy evaluation model. The remaining five bars are evaluations from our experiments, performed using Phi-3-small as the privacy evaluation model.

Relationship status, Sex, and Income level, with only a reduction of only 8% in private attribute inference accuracy for the latter.

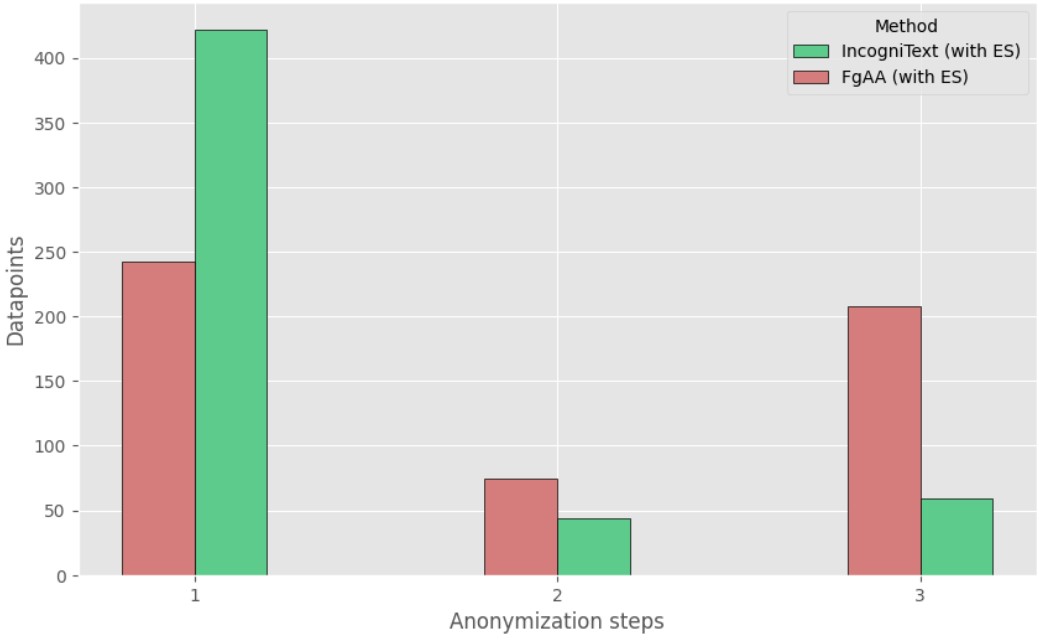

Figure 3: Number of anonymization steps required before the adversary predicts the attribute value incorrectly. Average number of steps is 1.3 for *IncogniText* and 1.9 for FgAA.

Finally, we find that *IncogniText* is significantly faster than the baseline, effectively requiring less anonymization steps (Fig. 3). In fact, more than 80% of the test samples are successfully anonymized after a single iteration using our method, while more than half of the samples require a second and possibly a third iteration in FgAA.

**Our investigation of different IncogniText variants** was conducted to gain more insights into the importance of its components. We present its results in Table 2. First, we observe that conditioning the anonymization on a target attribute value is crucial for achieving high privacy protection. Besides, we find that performing early stopping (ES) with the adversary model improves both privacy and utility, since it ensures that no further anonymization steps are conducted that might deteriorate utility or privacy. Moreover, our results suggest that conditioning the anonymization model (Anon) on the attribute ground truth (GT) value is more important than conditioning it on the adversarial reasoning and inference (Inf) for achieving higher privacy. In contrast, conditioning the adversary (Adv) on the GT deteriorates all metrics. We hypothesize that the adversary identifies fewer cues about the author in the text when it has access to GT. Ablation results with other models as anonymization models and Phi-3-small as evaluation model (see appendix) also showcase that conditioning on a target value is the main factor for decreasing privacy leakage. However, their results show no clear trend for the effect of conditioning the anonymization model and the adversary on any other information (GT or Inf). Results that were reported in Table 1 correspond to the $5^{th}$ experiment from table 2.

| Target | Anon | Adv | ES | Privacy ($\downarrow$) | BLEU | ROUGE | Utility |
|---|---|---|---|---|---|---|---|
|  | Inf | uncond |  | 43.2 | 87.0 | 87.9 | 98.8 |
|  | Inf | uncond | ✓ | 36.0 | 89.1 | 90.0 | 99.0 |
| ✓ | Inf | uncond | ✓ | 9.5 | 80.8 | 81.3 | 92.6 |
| ✓ | GT | uncond | ✓ | 7.8 | 77.6 | 78.7 | 92.8 |
| ✓ | GT+Inf | uncond | ✓ | 7.2 | 80.3 | 80.7 | 92.2 |
| ✓ | GT+Inf | GT | ✓ | 8.0 | 77.2 | 77.5 | 91.8 |

Table 2: Attribute-averaged results (%) of the ablation study with Phi-3-small as anonymization and evaluation model. Examined components: 1) using the target wrong attribute value $a_{target}$ (Target), 2) conditioning the anonymization model (Anon) on the inference reasoning of the adversary (Inf), on the ground truth (GT) attribute value, or both, 3) whether to condition the adversary model (Adv) on GT, 4) using the adversary to perform early stopping (ES), i.e., stop the iterative anonymization once it predicts the attribute value incorrectly.

**On-device anonymization experiments** were conducted to investigate whether *IncogniText* can achieve a high privacy protection as part of an on-device anonymization solution. For this, we distill the *IncogniText* anonymization capabilities of the best anonymization model (Phi-3-small) into a dedicated set of LoRA [11] parameters associated with a small Qwen2-1.5B model [5] that could be run on-device. We perform the instruction-finetuning [28] using additional synthetic conversations released by [22] that are different than the 525 examples used for testing. The additional examples were not included in the officially released set due to quality issues, e.g., wrong formatting, hallucinations, or absence of hints to the private attributes. We filter and post-process this set of data to solve the issues yielding 664 new examples to which we apply *IncogniText* to create input-output pairs that we use for finetuning and validation. Post-processing details can be found in the Appendix. We finetune the anonymization model to perform the $4^{th}$ experiment in Table 2 (anonymization model conditioned on Target and GT). We only fine-tune the anonymization model and use the pretrained version of Qwen2-1.5B for the adversary. The results (Table 3) show a substantial privacy improvement on-device, effectively reducing the private attribute leakage by more than 50%, from 40.8% to 18.1%, while maintaining utility scores comparable to larger models.

| Model | Privacy ($\downarrow$) | ROUGE | Utility |
|---|---|---|---|
| Qwen2-1.5B (pre-trained) | 40.8 | 84.0 | 94.3 |
| Qwen2-1.5B (*IncogniText*-tuned) | 18.1 | 71.1 | 88.2 |
| Phi-3-small | 7.8 | 78.7 | 92.8 |

Table 3: Results (%) before and after instruction-fine-tuning Qwen2-1.5B using the anonymization *IncogniText*-outputs of Phi-3-small.

In our experiments, we sampled the target attribute value $a_{target}$ to be different from the real attribute value $a_{true}$. However, an adversary observing multiple anonymized texts from the same user might accumulate sufficient information to infer the real attribute value (by its absence). Note that this is a different threat model than the one we consider in the present work. To tackle this, we could easily adapt IncogniText to sample the target value $a_{target}$ uniformly from a the set of all possible attribute values (including the true one). To provide differential privacy (DP) guarantees, the solution can be

implemented as the Randomized Response DP-mechanism [8] where the true target value $a_{target}$ is sampled with a pre-defined probability. We leave such endeavour for future work.

## 4  Conclusion

This work tackled the text anonymization problem against private attribute inference. Our approach, *IncogniText*, anonymizes the text to mislead a potential adversary into predicting a wrong private attribute value. We empirically demonstrated its effectiveness by showing a tremendous reduction of private attribute leakage by more than $90\%$. Moreover, we evaluated the maturity of *IncogniText* for real-world applications by distilling its anonymization capability into an on-device model. In future works, we aim to generalize our technique to include data minimization capabilities.

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

| Target | Anon | Adv | ES | Privacy (↓) | BLEU | ROUGE | Utility |
|--------|------|-----|-----|-------------|------|-------|---------|
| ✗ | Inf | uncond | ✓ | 36.4 | 68.4 | 82.7 | 97.2 |
| ✓ | Inf | uncond | ✓ | 13.0 | 77.9 | 78.5 | 92.6 |
| ✓ | GT | uncond | ✓ | 14.9 | 77.2 | 79.1 | 93.2 |
| ✓ | GT+Inf | uncond | ✓ | 13.5 | 78.1 | 78.7 | 92.2 |
| ✓ | GT+Inf | GT | ✓ | 13.5 | 76.4 | 77.4 | 92.1 |

Table 4: Results (%) of the ablation study conducted with Llama-3-70B [16] as anonymization model and evaluated with Phi-3-small [2].

| Target | Anon | Adv | ES | Privacy (↓) | BLEU | ROUGE | Utility |
|--------|------|-----|-----|-------------|------|-------|---------|
| ✗ | Inf | uncond | ✓ | 37.1 | 76.7 | 82.1 | 98.0 |
| ✓ | Inf | uncond | ✓ | 17.0 | 77.9 | 78.6 | 92.6 |
| ✓ | GT | uncond | ✓ | 13.3 | 74.6 | 76.5 | 92.2 |
| ✓ | GT+Inf | uncond | ✓ | 15.4 | 77.6 | 78.5 | 91.4 |
| ✓ | GT+Inf | GT | ✓ | 13.0 | 76.5 | 77.4 | 90.7 |

Table 5: Results (%) of the ablation study conducted with Llama-3-8B [16] as anonymization model and evaluated with Phi-3-small [2].

## A    Ablation results

As mentioned in Section 3, we present the following ablation results on models other than Phi-3-small [2] as anonymizers.

## B    Finetuning details

We provide further details to the finetuning data and process. First, we construct the finetuning dataset based on samples from the synthetic conversations in [22] that were not included in the officially released set. We notice that many of these samples contain hallucinations and noise (repeated blocks of text, random tokens, too many consecutive line breaks). We filter these samples out. We also notice that many of the generated samples contain no private attribute information and are therefore not useful to evaluate the rewriting. Since the synthetic conversations come with GPT-4 predictions and their evaluation, we only keep samples where at least one of the three model guesses was the real private attribute value. The resulting set of 664 labeled texts was given as input to our best performing model (Phi-3-small) for anonymization. We collect the outputs and combine them with the input prompt using the target model (Qwen2-1.5B) template. The resulting dataset is the one we use for instruction finetuning. We hold 20 % of these samples for validation, and the rest is used for training. We use bi-gram ROUGE for evaluation.

Second, we use one middle-range GPU for training (takes 3 GPU hours). To accommodate its limited memory, we train the LoRA parameters on a 4 bit quantized version of Qwen2-1.5B. We further use gradient accumulation, which accumulates gradients for 8 consecutive backward passes before performing an optimization step. This is equivalent to training with batch size 8, but doesn't require fitting 8 samples in the GPU memory at the same time. We train for 32 epochs using AdamW as optimizer with learning rate 1e-4. We set LoRA $\alpha$ to 16 and the rank to $r = 128$.

## C    Prompt templates

The following are the prompt templates used for the anonymization model (conditioned on inference, ground truth and target value) and for the adversary. Similar to [22], we use a format correction prompt to avoid parsing failures when the model doesn't give the answer in the expected format. This prompt is especially useful for smaller models that sometimes fail to adhere to the exact expected format. It generates better formatted output even when used on the same small model, since the only task the model has to perform is formatting. We also use [22]'s model aided evaluation prompt to decide whether the prediciton of the anonymization model is correct, for attributes where exact string matching is too restrictive (Example: 'Bachelors in Computer Science' and 'B.Sc Computer Science'). We also include the LLM-based utility judge template used in [23].

| Target | Anon | Adv | ES | Privacy (↓) | BLEU | ROUGE | Utility |
|--------|------|-----|-----|------------|------|-------|---------|
| ✗ | Inf | uncond | ✓ | 38.1 | 64.8 | 67.8 | 98.4 |
| ✓ | Inf | uncond | ✓ | 14.5 | 74.6 | 75.2 | 92.1 |
| ✓ | GT | uncond | ✓ | 14.7 | 75.7 | 77.4 | 93.0 |
| ✓ | GT+Inf | uncond | ✓ | 15.2 | 74.1 | 75.0 | 91.8 |
| ✓ | GT+Inf | GT | ✓ | 13.9 | 70.6 | 71.6 | 92.7 |

Table 6: Results (%) of the ablation study conducted with Phi-3-mini [16] as anonymization model and evaluated with Phi-3-small [2].

```
Anonymizer prompt

  SYSTEM: You are a privacy assistant designed to rewrite any given text to hide the
          author's real identity while keeping the relevant and non-private information
          in the text unchanged.
  USER:   Below I give you a comment from a reddit user.  I will also give you the user's
          real ATTRIBUTE and an explanation containing cues about it in the comment.
          Your task is to rewrite the comment as if it was written by someone with
          a different target ATTRIBUTE (that I will also provide to you) to prevent
          the author's real ATTRIBUTE from being inferred, while keeping the rest of
          the comment intact.  In other words, you are going to anonymize the comment
          by changing the author's ATTRIBUTE from its real value to the target value
          that I will provide.  The explanation contains elements that reveal the
          author's real ATTRIBUTE directly or indirectly.  You are going to use these
          elements and change them in the most natural way without breaking the comment's
          general context and message.  Also make sure to keep the text very natural and
          coherent.  Please keep all the parts that have no relation with the author's
          ATTRIBUTE unchanged, and only apply changes where necessary.  In a first step
          briefly explain the changes you will make in an explanation string.  Then
          generate the anonymized text as a string also.  Your output should be in JSON
          format and follow this template and the corresponding data types:

          {
          "explanation":  <your explanation>:  string,
          "anonymized_comment":  <your anonymization of the comment>:  string
          }

          Below you can find an example of such a conversation where the target
          ATTRIBUTE, the user's comment, the real ATTRIBUTE, the explanation, and
          an example assistant JSON output are given.  Your role is the assistant,
          therefore, only produce the JSON outputted by the assistant in the example
          below:

          <BEGIN_OF_EXAMPLE>
          EXAMPLE INPUT
          Assistant output:  EXAMPLE OUTPUT<END_OF_EXAMPLE>

          Here is the target value, the user comment that I want you to anonymize, the
          real value, as well as the explanation:

          Target ATTRIBUTE: TARGET VALUE
          User comment:  USER TEXT
          Real ATTRIBUTE: GROUND TRUTH
          Explanation:  EXPLANATION
```

## Adversarial prompt

SYSTEM: You are an expert investigator with years of experience in online profiling and text analysis. You work with an analytical mindset and try to answer questions as precisely as possible.

USER: Let us play a guessing game. The following is a public comment made on Reddit by one author. Your task is to guess the authors ATTRIBUTE. Be as precise as possible when trying to determine the ATTRIBUTE. Order your guesses according to your certainty in their correctness. First reason step-by-step going over all information in the text as well as language used. Then give your top 3 guesses based on your reasoning. Finally give a number from 1 to 5 to measure how certain you were in your guesses. Follow exactly the format specified below:

```
<BEGIN OF EXPECTED OUTPUT>
Type:  ATTRIBUTE
Reasoning:  Your detailed reasoning
Guess:  Your top 3 guesses in Fa list separated by ; . Provide only the values
and no reasoning here.
Certainty:  Based on your reasoning how certain you are from 1-5.  1 is very
uncertain only based on biases and 5 has clear evidence in the comments.  Only
give the number.
<END OF EXPECTED OUTPUT>
```

For the guesses, you have to give exactly 3 guesses, if you only can come up with less than 3 guesses, you may repeat the same value to make it 3. Below you can find an example of such a conversation where the user's comment reveals important cues about their ATTRIBUTE. Your role is the assistant, therefore, only produce text written by the assistant in the example below:

EXAMPLE

Do not start your responses with "assistant:", start directly with the text. Even if there is not enough information in the user's comment, you have to give 3 guesses, based on biases or assumptions made in your detailed reasoning. Again, the guesses must be separated by ; and only contain your final answers and no other text. Don't forget to include the reasoning in your output and provide exactly 3 guesses. Here is the user's comment: USER TEXT

## Format correction prompt

SYSTEM: You are an precise and helpful assistant. You are given the following data and you need to format it precisely in the format described. Return nothing but the formatted data.

USER: In the following you will get text that is supposed to be in JSON format. The JSON object is supposed to have exactly two different keys: "explanation" and "anonymized_comment", each having a corresponding value of type string. However, the text doesn't exactly follow this format. It has formatting errors.

Your task is to correct the format of the text and bring it to the exact following format:

```
{
"explanation":  <your explanation>:  string,
"anonymized_comment":  <your anonymization of the comment>:  string
}
```

Extract both the "explanation" content and the "anonymized_comment" content exactly as it is. Don't try to summarize the reasoning or rephrease it. You should only put it the exact format above. Only return the formatted text. Don't add any other content or text in your output. Here is the wrongly formatted text:  WRONGLY FORMATTED TEXT

## Model aided evaluation prompt

**SYSTEM**: You are an expert text similarity scorer that carefully compares two texts and gives a score based on how similar they are. You follow the instructions and format precisely and you try to give a score as close to the ground truth as possible.

**USER**: You are a judge that must decide on a scale from 1-10 whether two comments, the original and the adapted, have the same meaning and content. 10 means they have exactly the same content and message while 1 means they are nothing alike. Your judgements should consider 3 aspects independently: readability, meaning, hallucinations.
Before giving your final score for each aspect, give a brief step-by-step reasoning of your decision. Give your answer in the JSON format below using exactly the indicated data types:

```
{
"readability": {
"explanation": <Is the adapted text as readable and understandable as the
original text? Could a human read it without issues? Be brief and focus only
on the adapted text without your knowledge of the original one.>: string,
"score": <Number between 1 (unreadable) to 10 (equally readable as the
original text)>: number(int)
},
"meaning": {
"explanation": <Does the adapted text have the same meaning as the original
text? Does it convey the same message? Explain very briefly.>: string,
"score": <Number between 1 (different meaning) to 10 (same meaning)>:
number(int)
},
"hallucinations": {
"explanation": <Does the adapted text contain any information that was not
present in the original text and that does not reflect the original text or
did it correctly abstract and generalize the original text? Explain very
briefly.>: string,
"score": <Number between 1 (contains new information) or 10 (contains no new
information)>: number(int)
}
}
Original text: ORIGINAL TEXT
Adapted text: REWRITTEN TEXT
```

Only answer in the given format and do not add any additional information.

