# OpenReview forum: "IncogniText: Privacy-enhancing Conditional Text Anonymization via LLM-based Private Attribute Randomization"
_NeurIPS.cc/2024/Workshop/SafeGenAi — SafeGenAi Poster_

### Official Review · Reviewer_NW5e · 2024-10-09
**Reviewer NW5e**

**Rating:** 6
**Confidence:** 3

**Review:**

### Paper Summary

This paper introduces "IncogniText," a novel text anonymization technique designed to prevent adversaries from correctly inferring authors' private attributes like age, gender, income, etc., while maintaining the practicality of the text (meaning and semantics) as much as possible. The authors use a method based on large language models (LLMs) to transform original texts into anonymized versions, misleading potential privacy attackers into predicting incorrect target attribute values. Experimental evaluations demonstrate the technique's effectiveness on two datasets, showing significant reduction in private attribute leakage and superior protection compared to existing methods. Additionally, the paper distills IncogniText's anonymization capabilities into a set of LoRA parameters that can be integrated into small device models for consumer products.

### Strengths

1. **Innovation:** IncogniText introduces a unique approach using LLMs for conditional text anonymization, enhancing text privacy while retaining semantic integrity.
2. **Practicality:** The method can be integrated into practical applications, such as incorporating LoRA parameters into device models, enabling efficient text anonymization on endpoint devices.
3. **Thorough Experimental Evaluation:** The paper demonstrates IncogniText's superiority across multiple datasets compared to other methods, effectively proving its applicability and effectiveness in real-world scenarios.

### Weakness

1. **Insufficient Discussion on Utility Loss:** Although the paper mentions a focus on maintaining text utility (semantic integrity), it lacks detailed discussions on the utility losses post-anonymization, such as readability and information richness. This is key to fully understanding the impacts of anonymization.
2. **Unproven Generalization Capability:** The study focuses mainly on specific private attributes and datasets, suggesting a need for further research to confirm the method's effectiveness across a broader range of text types and scenarios.
3. **Inadequate Security Considerations:** While the anonymization results are impressive, the paper lacks sufficient discussion on defenses against potential reverse engineering or other forms of attacks, such as model inversion attacks to infer original data or attributes.

### Suggestions for Improvement

1. **Deepen Analysis of Utility Loss:** There should be a detailed analysis of potential utility losses in various aspects (e.g., language fluency, information integrity) post-anonymization. This could be achieved through qualitative assessments (like expert reviews) and quantitative methods (such as natural language understanding tests).
2. **Enhance Generalization Research:** It's recommended to test different types of text content and more private attributes to verify the method's generalization capabilities. Consider using diversified datasets and real-world application scenarios for further experiments.
3. **Strengthen Security Assessments:** Propose and implement specific security tests, such as simulating attacks to assess the resistance of anonymized texts to reverse engineering. Also, explore incorporating more complex obfuscation techniques or adding randomness to enhance overall security.

With these improvements, the paper could more comprehensively evaluate the real-world effectiveness of its method, while enhancing its practicality and security across various environments.

---

### Official Review · Reviewer_4PEK · 2024-10-09
**Attribute Anonymization in Text**

**Rating:** 7
**Confidence:** 5

**Review:**

This IncogniText paper investigates an interesting question on anonymizing the text without revealing the sensitive user attribute value. IncogniText misleads a potential attacker into predicting a wrong private attribute value according to the anonymization template and a target attribute value. The paper then demonstrates the maturity of our method for real-world applications by distilling its anonymization capability into a set of LoRA parameters. The experiments support their major findings.

---

### Official Review · Reviewer_sPY6 · 2024-10-09
**Review of Submission 133**

**Rating:** 6
**Confidence:** 4

**Review:**

This manuscript proposes a new method for anonymizing human users’ texts to address ever increasing AI-related privacy concerns. The proposed model, IncogniText, is interesting in that it takes a zero-shot approach (i.e., no training required) and enhances its performance through an iterative strategy inspired by adversarial training. While I believe this work does not meet the typical standard for a full paper, it presents intriguing work-in-progress results and could lead to engaging discussions at the workshop. Therefore, I recommend accept.

Below are my specific comments. Please note that I am not an expert on this very topic, so my feedback does not address the plausibility of the authors' choice of external baseline models (which is important!)

<Major concerns>

1.	L99-100 states that multiple attribute anonymization can be easily achieved by modifying the prompt for both the adversary and anonymization model. While I agree that this is straightforward to implement, my concern lies with the performance (in terms of both model effectiveness and computation time). Although anonymizer and adversary LLMs may perform adequately for a simple task (e.g., single-attribute setting), their performance might degrade when handling multiple attributes. Furthermore, in real-world applications, texts often contain multiple privacy-related attributes. Therefore, demonstrating whether the model’s performance scales effectively with multiple attributes is crucial.

2.	L58 suggests that a specific private attribute is provided to the model (IncogniText), rather than being identified by the model itself. This limits its practicality for real-world applications. While there may be instances where users wish to specify a particular private attribute for anonymization, I believe that most users would expect IncogniText to automatically identify private attributes and anonymize potential privacy concerns, without needing explicit input. Have you considered incorporating automatic detection of private attributes? Or, are there existing works that address this?

3.	Overall, I found the evaluation section lacking:
- Although ROUGE scores are presented in the tables (e.g., Tables 1-3), they are not discussed at all. If there is no intention to discuss these results, consider removing them.
- Ideally, the evaluation section (L161-217) should discuss IncogniText’s performance across multiple metrics. For example, what does it mean for ALS (one of the baseline methods) to have a worse Privacy score but a better ROUGE score, compared to IncogniText
- Regarding "Utility," which is an average of various numeric scores assigned by an LLM judge, my personal experience with both closed- and open-source LLMs suggests that LLMs are unreliable in providing numeric scores, even with detailed guidelines. While this evaluation method is gaining popularity, I have not encountered thorough research on an LLM's ability to generate reliable numeric scores. I’d like to see justification for using this LLM-judge-based metric.

<Minor concerns>

- The algorithm box on Page 3 should specify "private attribute" as a required input (see my comment #2).

- Unless the LLM is hosted locally (which would likely be the case, e.g., on a phone), there is a risk that the LLM chat (querying) history could remain on a centralized server, which could lead to further privacy data leakage.

---

### Official Review · Reviewer_H1EH · 2024-10-09
**The authors propose anonymization technique IncogniText that prevents adversaries from inferring private author attributes while maintaining text utility by misleading adversaries.**

**Rating:** 4
**Confidence:** 3

**Review:**

Pros:
- The proposed method, IncogniText, introduces a way of anonymizing text through attribute obfuscation rather than simply removing or masking identifiable data.
- The authors show that this method significantly reduces private attribute leakage.  IncogniText appears to be model-agnostic.

Conses/weaknesses:
- The authors acknowledge that an adversary could infer the original attributes by analyzing multiple anonymized texts over time, but no experiments and results were provided. Moreover, it simplifies the problem but overlooks more complex scenarios involving multi-source or time-based.
- lack of context dependence, e.g., There may be overlapping definitions of PII types across countries. For example, a UK passport number has 9 digits, similar to a phone number.
- The decision-making process for changing certain text elements to mislead attackers can introduce unintended meanings or hallucinations

The paper has some redundancy, particularly in the explanations of models and the iteration process. Several paragraphs repeat similar points, or even part of sentences, e.g., _a potential attacker into predicting a wrong private attribute value by
inserting_  (pages 3 and 4) or information such as that the authors use different models for the adversarial and attacker models (last 3 lines of p.2 and top 3 lines of p.3)

Suggestion: It would be beneficial to include metrics such as model losses or perplexities to evaluate the effectiveness of the fine-tuning process (to fully understand the results)